

# Combining citizen science, phylogenetics, and bioacoustics to inform taxonomy and conservation of the Near Threatened *Proceratophrys paviotii* (Anura, Odontophrynidae)

João Victor Andrade Lacerda[1], Diego J. Santana[2], Carla Guimarães[2], Alice Zanoni dos Santos[3], Alan P. Araujo[4], Natalia Pirani Ghilardi-Lopes[5] and Sarah Mângia[2]

[1] National Institute of the Atlantic Forest, Santa Teresa, Espirito Santo, Brazil
[2] Mapinguari Lab, Instituto de Biociências, Universidade Federal de Mato Grosso do Sul, Campo Grande, Mato Grosso do Sul, Brazil
[3] Instituto Federal do Espírito Santo campus Santa Teresa, Santa Teresa, Espírito Santo, Brazil
[4] Laboratório de Herpetologia e Comportamento Animal, Departamento de Ecologia, Instituto de Ciências Biológicas, Universidade Federal de Goiás, Campus Samambaia, Goiânia, Goiás, Brazil
[5] Center for Natural and Human Sciences, Federal University of ABC (UFABC), São Bernardo do Campo, São Paulo, Brazil

Corresponding author
João Victor Andrade Lacerda,
lacerdajva@gmail.com

## ABSTRACT

Herein, basel on novel data gathered by citizens scientists and specialists, we contribute to the improvement of scientific knowledge and conservation of the Near Threatened *Proceratophrys paviotii* in order to: 1) test for the first time the phylogenetic position and a species delimitation of *P. paviotii* through a molecular approach; 2) describe a larger sample of its advertisement call to properly encompass the species intraspecific variation; 3) describe for the first time the *P. paviotii* release call; and 4) provide novel insights on the species conservation status. Our 16S tree confidently grouped *P. paviotii* with *P. cururu*, *P. renalis*, and *P. laticeps*. The average sequence divergence between *P. paviotii* and its congeners ranged from 2.2% (*P. laticeps*) to 9.1% (*P. redacta*). Advertisement calls consisted of a single note with duration of 0.26–0.58 s, 17–41 pulses emitted at rate of 54.19–77.49 pulses/s and peak frequency of 775.19–947.46 Hz. Release calls consisted of a single note with duration of 0.04–0.43 s, 2–13 pulses emitted at rate of 21.17–81.58 pulses/s and peak frequency of 689.1–1,722.6 Hz. Additionally, our study strongly supports the notion that Citizen Science approaches can yield invaluable information concerning species' geographic distribution and conservation.

# INTRODUCTION

Amphibians are among the most endangered animal groups on the planet, with 41% of their species classified as threatened with extinction, including many Neotropical lineages under some risk (*Luedtke et al., 2023*). The Neotropical genus *Proceratophrys*

Miranda-Ribeiro, 1920 occurs in Brazil, Argentina, Paraguay and possibly Bolivia, and currently includes 43 species (*Frost, 2024*). Eight species of *Proceratophrys* are threatened with extinction, classified as Critically Endangered (CR), Vulnerable (VU), or Endangered (EN). Two species of *Proceratophrys* are classified as Near Threatened (NT). A taxon is classified as Near Threatened (NT) when it is close to or likely to qualify for a threatened category in the near future (*IUCN, 2024*). Therefore, these species require special attention and focus from both the scientific community and conservation initiatives.

Evolutionary studies have played an important role in anuran taxonomy. Several populations recovered as independent evolutionary units have supported the description of new taxa (*e.g.*, *Ferreira et al., 2023*; *Folly et al., 2024*). Conversely, taxa that have not been recovered as reciprocally monophyletic have led to synonymizations (*e.g.*, *Faivovich et al., 2021*; *Pereira et al., 2022*). Among *Proceratophrys*, the same scenarios are observed with species that have been recently described (*Santana et al., 2021a*, *2021b*; *Mângia et al., 2022*) and synonymized (*Mângia et al., 2020*) supported by phylogenetic approaches. These studies have great potential to impact conservation status, given that we protect nominal species.

Citizen Science (CS) approaches have great potential in providing information on several biological groups, such as fungi, plants and animals, even on data deficient, rare, or threatened species (*e.g.*, *Mori & Menchetti, 2014*; *Campanaro et al., 2017*; *Irga, Barker & Torpy, 2018*; *Heard, Chen & Wen, 2019*; *Pirotta et al., 2020*; *Deacon, Govender & Samways, 2023*; *Farquhar et al., 2023*; *Krueger et al., 2023*; *Lacerda et al., 2023*), once such projects have encouraged the participation of the public in different stages of scientific projects, such as gathering and/or analyzing biological data (*Bonney et al., 2009*).

Citizen Science projects focused on anurans have contributed to a vast range of issues in science and conservation (*e.g.*, *Pittman & Dorcas, 2006*; *Price & Dorcas, 2011*; *Cosentino et al., 2014*; *Westgate et al., 2015*; *Sterrett et al., 2019*; *Antúnez-Fonseca et al., 2021*; *Ceríaco et al., 2021*; *Lee et al., 2021*; *Forti et al., 2022a*, *2022b*; *Glorioso et al., 2022*; *Forti & Szabo, 2023*). The public usually contributes by sending pictures and thus providing data on species occurrence *via* digital platforms, such as the iNaturalist (*e.g.*, *Forti et al., 2022a*, *2022b*; *Forti & Szabo, 2023*). This active involvement of the public in sharing visual data enables researchers to gain insights into the distribution and presence of anuran species. However, while the pictures are invaluable for assessing anuran's identity, CS projects often overlook a critical aspect: a vast vocal repertoire of this group, emitted depending on its social context, such as reproductive, aggressive, defensive, and feeding calls (*Köhler et al., 2017*). In anurans, advertisement calls are heritable and usually species-specific (*Duellman & Trueb, 1994*; *Wells, 2007*). Despite the importance of bioacoustics for taxonomic and conservation purposes, only a few CS projects focus on receiving anurans audio files from the public. For instance, the FrogID project gathers audio files containing anuran vocalization from Australian citizens (https://www.frogid.net.au/) (*Rowley et al., 2019*). This project has contributed to several scientific and conservation issues, such as: fire effect (*Rowley, Callaghan & Cornwell, 2020*), invasive species (*Rowley & Callaghan, 2022*), species geographic distribution (*Cutajar et al., 2022*), behavior (*Thompson et al., 2022*), urban impact (*Liu et al., 2022*), among others.

Since 2020, we have been conducting a CS project entitled *Cantoria de Quintal* (translated as "Songs from the Back Yard") in the municipality of Santa Teresa (*Lacerda et al., 2023*; *Lacerda, Santos & Lima, 2023*). We have encouraged the public to record anurans from their yards and/or surroundings using smartphones. The *Cantoria de Quintal* project aims to investigate scientific and conservation issues, such as species distribution, whether threatened species also occur in unprotected areas, urban and deforestation impacts, among others. However, species identification through audio files is not always easy and can be hampered when dealing with rare or poorly known species.

During 2020–2023 we received 42 audio files as part of the *Cantoria de Quintal* project from 10 citizen scientists. These files contained calls that we initially categorized as belonging to *Proceratophrys* sp.. After assessing literature information (*Cruz, Prado & Izecksohn, 2005*; *Ferreira et al., 2019a*), we assigned those records to *Proceratophrys* sp. (cf. *paviotii*). We were unable to confirm its identification to the species level due to two main reasons: 1) the vocalization of *P. paviotii* is not available for comparison in any existing public audio collections; and 2) the call description provided by *Cruz, Prado & Izecksohn (2005)* in the species' original description is based on only seven calls from a single male, which certainly do not properly encompass the species' intra-specific variation. Motivated by the challenge of properly identifying the citizen scientists' records, we initiated an integrative investigation on the *Proceratophrys paviotii* taxonomy.

*Proceratophrys paviotii* (Fig. 1A) is a Near Threatened species (*IUCN, 2024*) that occurs in the central and northern State of Espírito Santo (*Prado & Pombal, 2008*; *Almeida, Gasparini & Peloso, 2011*; *Peres & Simon, 2011*; Figs. 1B–1D). This species has never been included in phylogenetic or molecular species delimitation approaches so far (see *Mângia et al., 2020*; *Santana et al., 2021a*, *2021b*; *Mângia et al., 2022*). Therefore, its phylogenetic position has not been properly tested.

Efficient conservation predictions and strategies depend on filling knowledge gaps such as on its taxonomic identity (Linnean shortfall), geographic distribution (Wallacean shortfall), and evolution (Darwinian shortfall) (see *Hortal et al., 2015*). Therefore, combining data provided by citizen scientists and specialists, in this study we: 1) test for the first time the phylogenetic position and a species delimitation of *Proceratophrys paviotii* through a molecular approach; 2) describe a larger sample of its advertisement call to properly encompass the species intraspecific variation; 3) describe for the first time the *P. paviotii* release call; and 4) provide novel occurrence data and insights on the species conservation status. Thus, herein we contribute to the improvement of scientific knowledge and conservation of the Near Threatened *Proceratophrys paviotii*.

## MATERIALS AND METHODS

### Study area

The Municipality of Santa Teresa, State of Espírito Santo, located within the Atlantic Forest of Southeastern Brazil (Fig. 1), is known for its rich biodiversity across various biological groups (*e.g.*, *Thomaz & Monteiro, 1997*; *Brown & Freitas, 2000*; *Passamani, Mendes & Chiarello, 2000*; *Wendt et al., 2010*; *Gatti et al., 2014*; *Novaes et al., 2016*). With 109 amphibian species, Santa Teresa stands out for harboring one of the highest diversities

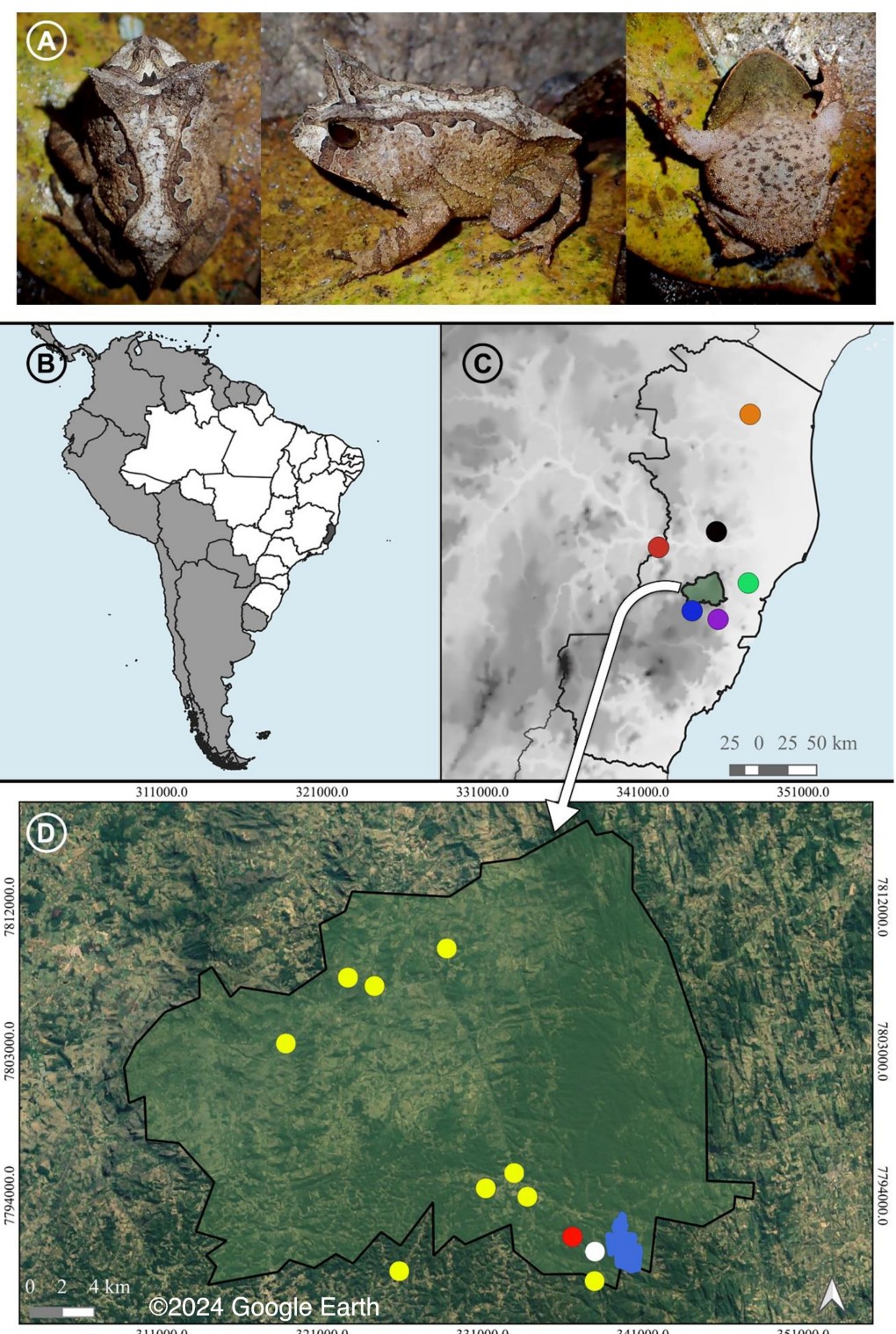

**Figure 1 A recorded male and the distribution of *Proceratophrys paviotii*.** (A) A recorded male of *Proceratophrys paviotii* (MBML 12369; snout-vent-length, SVL 44.1 mm): dorsal, lateral and ventral views. (B) State of Espírito Santo (shaded) in Southeastern Brazil. (C) Municipality of Santa Teresa (shaded) and the known distribution of *Proceratophrys paviotii* in the State of Espírito Santo: Municipality of Aracruz (green circle; *Prado & Pombal (2008)*), Municipality of Baixo Guandu (red circle; *Prado & Pombal (2008)*), Municipality of Santa Leopoldina (purple circle; *Prado & Pombal (2008)*),

**Figure 1** (continued)
Municipality of Santa Maria do Jetibá (blue circle; *Almeida, Gasparini & Peloso (2011)*), Municipality of Marilândia (black circle; *Almeida, Gasparini & Peloso (2011)*), and Municipality of Pinheiros (red circle; *Peres & Simon (2011)*). (D) Municipality of Santa Teresa, type locality of *P. paviotii* Estação Biológica Santa Lucia (blue); records of *P. paviotii* made through audio files sent by citizen scientists (yellow); records of *P. paviotii* made by formal scientists while visiting a citizen scientists' property (red); and records of *P. paviotii* made by formal scientists (white). Map built using the Google Satellite tool from QGis 3.22.2: source IBGE (https://www.ibge.gov.br/geociencias/organizacao-do-territorio/malhas-territoriais/15774-malhas.html). © 2024 Google Earth.

of this group in the world (*Ferreira et al., 2019a*; *Lacerda et al., 2021*). Notably, Santa Teresa is the only place where six species of *Proceratophrys* can be found in sympatry (*Prado & Pombal, 2008*; *Ferreira et al., 2019a*): *Proceratophrys boiei* (Wied-Neuwied, 1824); *P. laticeps* Izecksohn & Peixoto, 1981; *P. moehringi* *Weygoldt & Peixoto, 1985*; *P. paviotii* *Cruz, Prado & Izecksohn, 2005*; *P. phyllostomus* *Izecksohn, Cruz & Peixoto, 1998*; and *P. schirchi* Miranda-Ribeiro, 1937.

## Phylogenetic inference and genetic distances

We extracted DNA from tissue samples (liver) using the QIAGEN DNeasy Blood and Tissue Kit (Valencia, California, USA) following the manufacturer's protocol. Next, we amplified a fragment of the mitochondrial 16S gene using primers 16Sar and 16Sbr (*Palumbi et al., 1991*). The PCR protocol was configured with one initial phase of 94 °C for 3 min, followed by 35 cycles of 94 °C for 20 s, 50 °C for 20 s, 72 °C for 40 s, and a final extension phase of 72 °C for 5 min. Purification of PCR products and sequencing were performed by ACGTene Análises Moleculares Ltda. (Alvorada, Rio Grande do Sul, Brazil).

We combined our newly generated 16S sequences with comparable 16S sequences of *Proceratophrys* individuals available on GenBank, and also included the outgroups *Odontophrynus* spp., *Macrogenioglottus alipioi*, *Cycloramphus acangatan* and *Thoropa miliaris*, which are available in GenBank (Document S1). We aligned 16S mtDNA gene fragments using the MAFFT algorithm (*Katoh & Toh, 2008*) in Geneious v9.0.5 with default settings. The final dataset comprised 84 sequences of a 498 base-pair (bp) fragment of the 16S gene. All GenBank accession numbers and genetic voucher samples used here are listed in the Supplemental Material. We used the Bayesian Information Criterion in jModelTest (*Darriba et al., 2012*) to determine that GTR+I+G was the best model of nucleotide substitution for our 16S data set. To estimate phylogenetic relationships, we used Bayesian Inference (BI) in BEAST v2.6.6 (*Bouckaert et al., 2019*) for 20 million generations, sampling every 2,000, using a Yule speciation prior, implementing a relaxed clock log normal. We checked for stationarity by visually inspecting trace plots and ensuring that all effective sample sizes were above 200 in Tracer v1.7.1 (*Rambaut et al., 2018*). The first 10% of sampled genealogies were discarded as burn-in, and the maximum clade credibility tree with median node ages was calculated with TreeAnnotator v2.6.6 (*Bouckaert et al., 2019*). We calculated sequence divergences (uncorrected *p*-distances) among species/individuals using MEGA v10.1.1 (*Kumar et al., 2018*). In order to explore the relationship among haplotypes, we estimated haplotype networks among species closely related to the *P. paviotii* for the 16S mtDNA gene in POPART

**Table 1 Data on geographic occurrence of *Proceratophrys paviotii*.** Municipality, file voucher, coordinates, and area description. An asterisk (*) indicates files that were included in the bioacoustical analysis.

| Municipality | Voucher | Coordinates | Area description |
|---|---|---|---|
| Santa Teresa | FNJV 60391 | 19°56′20″S; 40°36′46″W | A wetland inside the urban area |
| | FNJV 60388* | 19°56′19″S; 40°36′45″W | A street located at the border of the urban area (rainwater stored by the sidewalk channel) |
| | FNJV 60397 | 19°56′38″S; 40°35′17″W | A vacant lot located at the border of the urban area |
| | FNJV 60396 | 19°49′10″S; 40°41′37″W | A yard located at a rural area |
| | FNJV 60389*, 60390*, 60392–60395 | 19°49′27″S; 40°40′40″W | A yard located at a rural area |
| | – | 19°48′03″S; 40°38′01″W | A yard located at a rural area |
| | – | 19°51′21″S; 40°43′39″W | A yard located at a rural area |
| | – | 19°55′54″S; 40°35′42″W | A street located at the border of the urban area |
| | FNJV 60377–60383* | 19°58′00″S; 40°33′41″W | A yard located at a rural area |
| | FNJV 60384–60387* | 19°58′30″S; 40°32′53″W | Rainwater stored at a roadside |
| Santa Leopoldina | – | 19°59′37″S; 40°32′41″W | A yard located at a forest edge |
| Santa Maria de Jetibá | – | 19°59′11″S; 40°39′58″W | A yard located at a rural area |

(*Leigh & Bryant, 2015*) using the median-joining network method. We identified each species using different colors in the haplotype network. All molecular protocols follow previous studies on *Proceratophrys* taxonomy (*e.g.*, *Santana et al., 2021a*, *2021b*; *Mângia et al., 2022*).

## Bioacoustics

Records (Table 1) made by specialists were performed with Tascam DR-40 and Tascam DR-05 recorders at a sampling rate of 44.1 kHz and 24 bits resolution (FNJV 60377–60383, 60384–60387). One male (MBML 12366) emitted release calls when handled (FNJV 60382). The citizen scientists' files were recorded using smartphones: *Motorola Moto G7 plus* (FNJV 60388, 60391), *Samsung Galaxy A21s* (FNJV 60397) and other two unspecified models (FNJV 60389–60390, 60392–60396). These files were sent *via Whatsapp* app (.ogg format) and then converted to *.wav* format. Bioacoustic analyses were performed using Raven Pro 1.5 software (*Bioacoustics Research Program, 2014*). Spectrograms were produced using Hann window type, DFT 256 sample size, and time grid overlap of 80.1%. Figures were built using seewave and tuneR packages in R (*R Core Team, 2017*), with window length = 512, overlap = 80.1%. Bioacoustic terminology follows the call-centered approach proposed by *Köhler et al. (2017)*. We calculated call duration (s), number of pulses per call (pulses/call), pulse emission rate per call (pulses/s), pulse duration (s), interval between pulses (s), peak frequency (Hz), frequency 5% (Hz), and frequency 95% (Hz). Quantitative parameters are presented as minimum–maximum (mean ± standard deviation; *n* = sample size). Low frequencies up to 200 Hz (safely below the minimum frequency reached by *P. paviotii*) were high-pass filtered to decrease background noise in the recording files using Raven Pro 1.5. Recordings were then compared to the literature available for other *Proceratophrys* species. Vocalization recordings were deposited at Fonoteca Neotropical Jacques Vielliard (FNJV; https://www2.ib.unicamp.br/fnjv/).

We examined the variability of acoustic parameters in male calls, focusing on both intra-individual and inter-individual levels. For assessing the intra-individual coefficient of variation (CVintra), we utilized a dataset of 105 calls from 11 individuals and computed the mean ($\bar{X}$) and standard deviation (SD) of individual calls using the formula CV = (SD/$\bar{X}$) × 100. To determine inter-individual variation (CVinter), we evaluated the mean and standard deviation of parameter values across all individuals, including two cases where only one call was available for each. Parameters with CV values ≤ 5% are considered static, while those with CV values ≥ 12% are considered dynamic (*Gerhardt, 1991*). We calculated both CVintra and CVinter for six parameters (note duration, peak frequency, 5% frequency, 95% frequency, pulse number, and pulses/second). Additionally, to discern whether intra-individual variation was greater than inter-individual variation, we computed the CVintra/CVinter ratio. If CVintra/CVinter is >1, it means a greater variation between males (inter-individual) than an intra-individual variation (*Gerhardt, 1991*; *Moser et al., 2022*).

## Occurrence data through Citizen Science and by specialists

We received 42 audio files (*.ogg* format) through the *Cantoria de Quintal* project containing calls emitted by *Proceratophrys paviotii*. Files were sent by eight citizen scientists from eight distinct localities in the Municipality of Santa Teresa, one in Santa Maria de Jetibá, and one in Santa Leopoldina, State of Espírito Santo (Fig. 1; Table 1). The *Cantoria de Quintal* activities include visiting citizen scientists to provide environmental education and enhance their engagement in the project. During one of these visits (October 23rd, 2020), we found a population of *Proceratophrys paviotii* inhabiting the citizen scientist's yard and surroundings with several males in calling activity (FNJV 60377–60383). This property (19°58′00″S; 40°33′41″W) is located about 2.2 km straight line from Estação Biológica de Santa Lúcia (type locality of *P. paviotii*; Fig. 1). Males were calling from the banks of an anthropized sandy stream. Recordings took place from 06:00 to 08:30 pm, during a light rain, air temperature ca. 20 °C. Additionally, we found a second population of *Proceratophrys paviotii* (February 19th, 2021) with several males calling from the rainwater stored at a roadside about 800 m from the species type locality (Fig. 1). We recorded five of these males (FNJV 60384–60387). Recordings took place from 9:00 to 10:00 pm, during a light rain, air temperature ca. 20 °C.

## Voucher specimens

Specimens collected as vouchers were anesthetized and killed with lidocaine 2%, fixed in formaldehyde 10%, and preserved in 70% ethanol at Museu de Biologia Prof. Mello Leitão (MBML) from Instituto Nacional da Mata Atlântica (INMA) (MBML 12366–12370). Before fixation, specimens had tissue (liver) samples extracted and stored in 96% ethanol for whole genomic DNA extraction. We have followed CONCEA (*Jared et al., 2023*) for guidelines for the ethical treatment. The Instituto Chico Mendes de Conservação da Biodiversidade (ICMBio #63575-5) and animal research ethics committee of the Universidade de Vila Velha (CEUAUVV #491-2018) provided sampling permits.

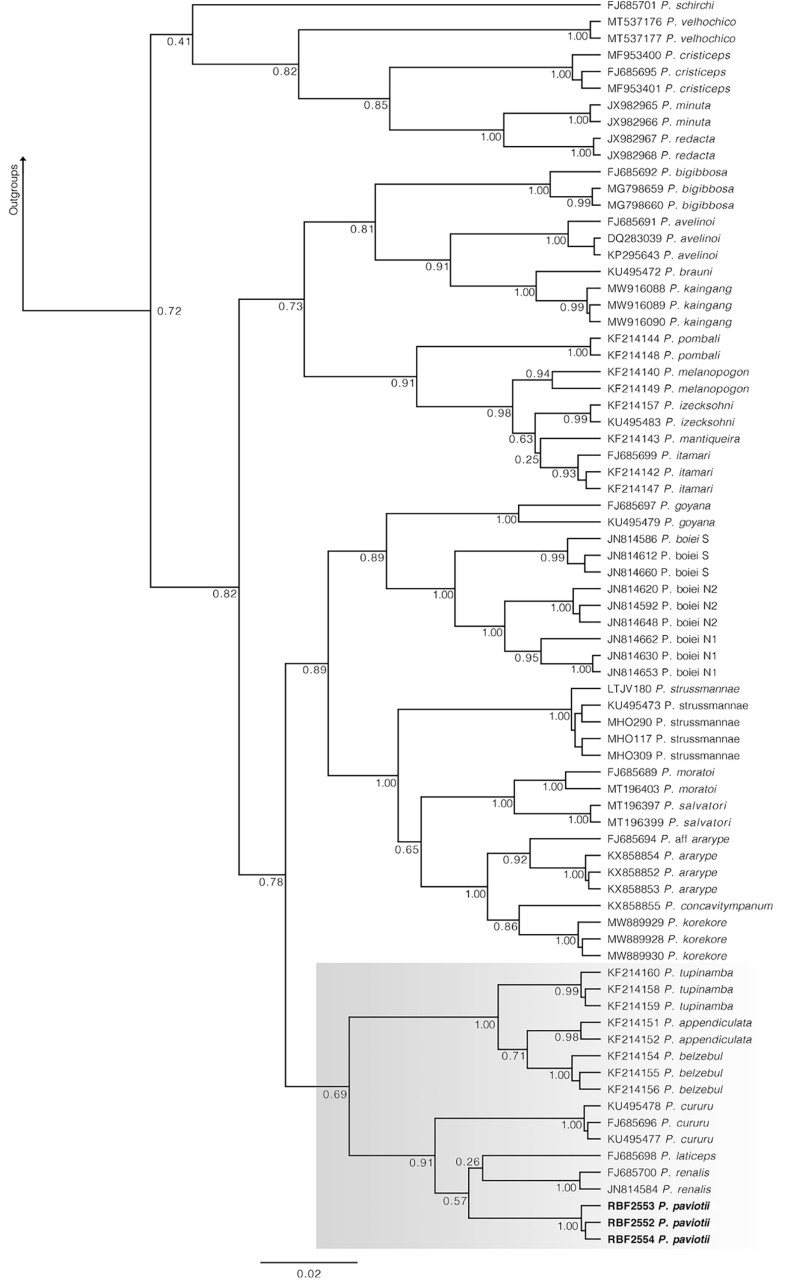

**Figure 2** **Phylogenetic analysis of the 16S mtDNA gene for the *Proceratophrys* spp.** Nodes are labeled with the Bayesian posterior probability. Scale bar is substitutions/site. Gray box indicates the lineage of *P. paviotii* and closely related species used on the haplotype network. DJ Santana prepared the tree figure using FigTree v1.4.4.

## RESULTS

### Phylogenetic inference and genetic distances

In our 16S tree analysis, *Proceratophrys paviotii* was grouped with *P. cururu*, *P. renalis*, and *P. laticeps* with a high posterior probability (PP = 0.91), forming a distinct clade (Fig. 2). Additionally, *P. appendiculata*, *P. belzebul*, and *P. tupinamba* formed a sister clade to this

**Table 2 Mean *p*-distance for mtDNA 16S of *Proceratophrys paviotii* to other species of the genus.**

| Species | *p*-distance | Species | *p*-distance |
|---|---|---|---|
| *P.* aff. *ararype* | 0.076 | *P. kaingang* | 0.051 |
| *P. appendiculata* | 0.028 | *P. korekore* | 0.077 |
| *P. ararype* | 0.081 | *P. laticeps* | 0.022 |
| *P. avelinoi* | 0.059 | *P. mantiqueira* | 0.038 |
| *P. belzebul* | 0.035 | *P. melanopogon* | 0.041 |
| *P. bigibbosa* | 0.046 | *P. minuta* | 0.080 |
| *P. boiei* N1 | 0.056 | *P. moratoi* | 0.068 |
| *P. boiei* N2 | 0.064 | *P. pombali* | 0.049 |
| *P. boiei* S | 0.037 | *P. redacta* | 0.091 |
| *P. brauni* | 0.060 | *P. renalis* | 0.030 |
| *P. concavitympanum* | 0.086 | *P. salvatori* | 0.067 |
| *P. cristiceps* | 0.087 | *P. schirchi* | 0.073 |
| *P. cururu* | 0.027 | *P. strussmannae* | 0.069 |
| *P. goyana* | 0.036 | *P. tupinamba* | 0.024 |
| *P. itamari* | 0.034 | *P. velhochico* | 0.082 |
| *P. izecksohni* | 0.033 | | |

group (PP = 1.00). However, the relationship between these clades was not strongly supported (PP = 0.69). Due to the utilization of a single mtDNA locus for species barcoding, our tree had low posterior probabilities for several nodes within *Proceratophrys*, which is expected. The average sequence divergence between *P. paviotii* and its congeners ranged from 2.2% (*P. laticeps*) to 9.1% (*P. redacta*) (Table 2). Furthermore, the mitochondrial haplotype network, based on a fragment of the 16S gene of *P. paviotii* and its closely related species (see Fig. 3), shows seven distinct mitochondrial lineages, with no haplotype sharing between them.

## Bioacoustics

We analyzed a total of 107 advertisement calls emitted by 13 males of *Proceratophrys paviotii* (Table 3). Calls consisted of a single note with duration of 0.26–0.58 s (0.46 s ± 0.07; *n* = 107 calls), 17–41 pulses (31.17 pulses ± 5.23; *n* = 104 calls), pulse duration of 0.007–0.026 s (0.012 s ± 0.002; *n* = 755 pulses), pulses emitted at rate of 54.19–77.49 pulses/s (68.32 pulses/s ± 6.64; *n* = 104 calls). Calls had peak frequency of 775.19–947.46 Hz (861.85 Hz ± 33.09; *n* = 101 calls), frequency 5% of 656.25–861.33 Hz (718.85 Hz ± 45.51; *n* = 101 calls), and frequency 95% of 937.5–1,119.73 Hz (1,001.91 Hz ± 54.84; *n* = 101 calls). Calls have an ascendant amplitude modulation on its beginning and descendant amplitude modulation on its ending (Fig. 4A).

We analyzed a total of 55 release calls emitted by a single male of *Proceratophrys paviotii*. Release calls consisted of a single note with duration of 0.04–0.43 s (0.15 s ± 0.07; *n* = 55 calls), 2–13 pulses (7.78 pulses ± 3.22; *n* = 55 calls) with 0.003–0.023 s (0.008 s ± 0.002; *n* = 427 pulses) emitted at rate of 21.17–81.58 pulses/s (51.56 s ± 11.9; *n* = 55 calls).
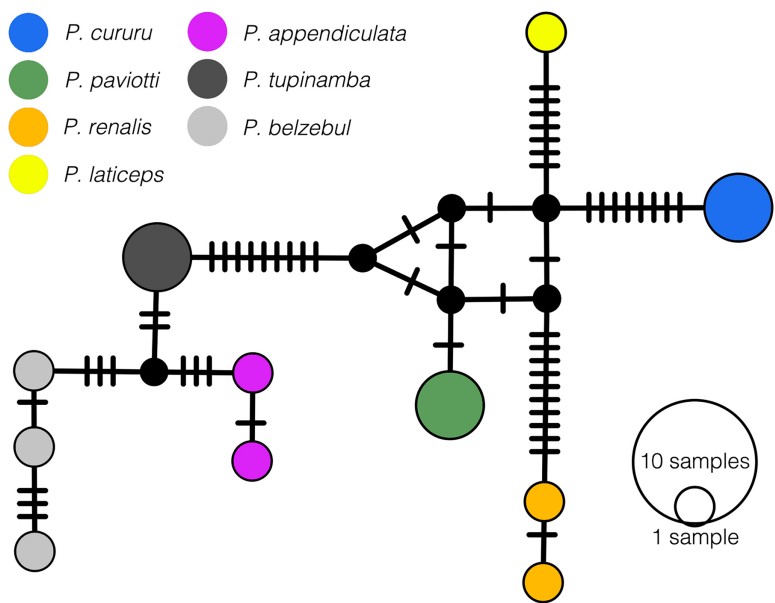

**Figure 3 Median-joining haplotype network based on mtDNA 16S of *Proceratophrys pavitotii*.** Each haplotype circle is proportional to its frequency (indicated in legend). Each color represents distinct species, and black dots represent inferred unsampled or extinct haplotypes. Mutational steps between alleles are represented by lines. *Proceratophrys cururu* (blue), *P. paviotii* (green), *P. renalis* (orange), *P. laticeps* (yellow), *P. appendiculata* (pink), *P. tupinamba* (dark grey), *P. belzebul* (light grey). DJ Santana prepared the Haplotype network using POPART.

Release calls had peak frequency of 689.1–1,722.6 Hz (1,227.8 Hz ± 424.3; *n* = 55 calls), frequency 5% of 516.8–861.3 Hz (667.1 Hz ± 81.6; *n* = 55 calls), and frequency 95% of 1,205.9–3,445.3 Hz (2,211.7 Hz ± 322.7; *n* = 55 calls). Release calls had irregular patterns of amplitude and frequency modulation. Isolated pulses were sporadically emitted between calls (Fig. 4B).

The CVintra demonstrated a static variation in peak frequency, 5% frequency, 95% frequency and pulse rate, and an intermediate variation in note duration and number of pulses per call. On the other hand, in a CVinter approach, note duration and number of pulses per call were both considered dynamic; pulse rate was considered intermediate; and peak frequency, 5% frequency, 95% frequency were considered static (Table 4). Additionally, the ratio between CVinter and CVintra was greater than one (>1) in all parameters, so it was considered a dynamic variation.

The advertisement call of *Proceratophrys paviotii* differs from those of *P. carranca*, *P. goyana*, *P. rotundipalpebra*, and *P. vielliardi* by its single note call pattern (multi noted call pattern in these species). The advertisement call of *P. paviotii* differs from those of *P. appendiculata*, *P. bigibbosa*, *P. boiei*, *P. brauni*, *P. cururu*, and *P. moehringi* by its shorter notes. The advertisement call of *P. paviotii* differs from those of *P. carranca* and *P. goyana* by its longer notes. The advertisement call of *P. paviotii* differs from those of *P. appendiculata* and *P. cristiceps* by its lower number of pulses per note. It differs from those of *P. bigibbosa*, *P. boiei*, *P. brauni*, *P. cururu*, *P. moheringi*, and *P. palustris* by its higher pulse emission rate. It differs from those of *P. ararype, P. carranca,*

**Table 3 Quantitative parameters (presented as range) of the advertisement call of *Proceratophrys* species.** When more than one source was available, we combined the values provided in a single range. Values in bold do not overlap with *P. paviotii*.

| Species | Notes/ call | Note duration (s) | Pulses/ note | Pulses/s | Dominant frequency (kHz) | Source |
|---|---|---|---|---|---|---|
| *P. paviotii* | 1 | 0.26–0.58 | 17–41 | 54.2–77.5 | 0.775–0.947 | Present study |
| *P. paviotii* | 1 | 0.34–0.43 | 26–32 | x | 0.660–1.280 | *Cruz, Prado & Izecksohn (2005)* |
| *P. appendiculata* | 1 | **1.32–2.41** | **51–129** | 30.0–65.4 | **0.562–0.656** | *Dias et al. (2013)* |
| *P. ararype* | 1 | 0.37–0.65 | 38–65 | **95.7–102.7** | 1.033–1.378 | *Mângia et al. (2018)* |
| *P. avelinoi* | 1 | 0.22–0.75 | 23–70 | 64.0–72.0 | 1.050–2.300 | *Kwet & Baldo (2003)* |
| *P. bigibbosa* | 1 | **1.60–1.90** | 40–45 | **23.0–27.0** | 1.050 | *Kwet & Faivovich (2001)* |
| *P. boiei* | 1 | **0.70–0.80** | 30–35 | **45.0** | 0.350–1.350 | *Heyer et al. (1990)* |
| *P. brauni* | 1 | **0.70–0.90** | 24–28 | **35.0–40.0** | **1.350** | *Kwet & Faivovich (2001)* |
| *P. carranca* | 1–10 | **0.04–0.19** | 5–21 | **95.2–131.5** | 1.033–1.378 | *Godinho et al. (2013)* |
| *P. concavitympanum* | 1 | 0.18–0.50 | 19–51 | **100.0–119.7** | **754.6–1,186.3** | *Santana et al. (2010)* |
| *P. cristiceps* | 1 | 0.52–0.79 | 33–69 | **78.5–91.8** | 0.860–1.030 | *Nunes & Juncá (2006)*, *Nunes et al. (2015)* |
| *P. cururu* | 1 | **1.20** | x | **45.0** | 0.600–1.000 | *Eterovick & Sazima (1998)* |
| *P. goyana* | 1–34 | **0.06–0.24** | 7–23 | **83.3–120.5** | 0.937–1.125 | *Martins & Giaretta (2013)* |
| *P. huntingtoni* | 1 | 0.2–0.3 | 19–25 | x | 1.095–1.3445 | *Ávila, Pansonato & Strüssmann (2012)* |
| *P. itamari* | 1 | 0.4–0.8 | 20–42 | 49.0–55.0 | 1.033–1.205 | *Mângia et al. (2014)* |
| *P. korekore* | 1 | 0.16–0.33 | 18–32 | **96.4–111.1** | 0.861 | *Santana et al. (2021b)* |
| *P. laticeps* | 1 | 0.49–1.57 | 28–94 | 44.0–74.4 | 0.340–0.680 | *Araújo et al. (2021)*, *Martins & Giaretta (2021)*, *Sichieri et al. (2021)* |
| *P. mantiqueira* | 1 | 0.17–0.48 | 12–41 | 68.0–96.0 | 0.999–1.274 | *Mângia, Santana & Feio (2010)* |
| *P. melanopogon* | 1 | 0.40–0.80 | 20–38 | 40.0–55.0 | 0.831–1.033 | *Mângia et al. (2014)* |
| *P. minuta* | 1 | 0.40–0.72 | 30–52 | 66.8–75.2 | **1.980–2.070** | *Nascimento et al. (2019)* |
| *P. moehringi* | 1 | **3.50–4.00** | x | **33.0–40.0** | 0.200–0.700 | *Weygoldt & Peixoto (1985)* |
| *P. moratoi* | 1 | 0.14–0.33 | 12–26 | 65.0–103.0 | 1.153–1.594 | *Brasileiro, Martins & Jim (2008)*, *Martins & Giaretta (2012)*, *Magalhães et al. (2020)* |
| *P. palustris* | 1 | 0.33–0.75 | 12–26 | **32.0–37.0** | **1.464** | *Martins & Giaretta (2012)* |
| *P. pombali* | 1 | 0.30–0.60 | 30–49 | 76.2–88.8 | **1.312–1.550** | *Malagoli, Mângia & Haddad (2016)* |
| *P. redacta* | 1 | 0.15–0.85 | 13–74 | **84.8–106.7** | **1.697–2.142** | *Simões et al. (2020)* |
| *P. renalis* | 1 | 0.15–0.46 | 13–30 | 61.5–86.1 | 0.689–1.033 | *Santana et al. (2011)* |
| *P. rotundipalpebra* | 1–24 | 0.04–0.33 | 4–32 | **78.1–130.4** | 1.125–1.453 | *Martins & Giaretta (2013)* |
| *P. salvatori* | 1 | 0.19–0.42 | 15–25 | 54.0–61.0 | **1.572–1.875** | *Bastos et al. (2011)*, *Magalhães et al. (2020)* |
| *P. sanctaritae* | 1 | 0.20–0.90 | 31–94 | **102.3–142.4** | 0.950–1.290 | *Cruz & Napoli (2010)* |
| *P. schirchi* | 1–6 | 0.18–0.45 | 15–31 | 66.0–93.4 | 0.861–1.810 | *Nascimento et al. (2019)*, *Sichieri et al. (2021)* |
| *P. velhochico* | 1 | 0.32–0.51 | 29–46 | **85.4–96.5** | **1,312.5** | *Mângia et al. (2022)* |
| *P. vielliardi* | **3–20** | 0.04–0.30 | 4–30 | **95.6–118.8** | 1.022–1.291 | *Martins & Giaretta (2011)* |

*P. concavitympanum, P. cristiceps, P. goyana, P. korekore, P. redacta, P. rotundipalpebra, P. sanctaritae, P. velhochico*, and *P. vieliardi* by its lower pulse emission rate. The advertisement call of *P. paviotii* differs from those of *P. brauni, P. minuta, P. moratoi, P. palustris, P. pombali, P. redacta, P. salvatori*, and *P. velhochico* by its lower peak of frequency. The advertisement call of *P. paviotii* differs from those of *P. appendiculata* and

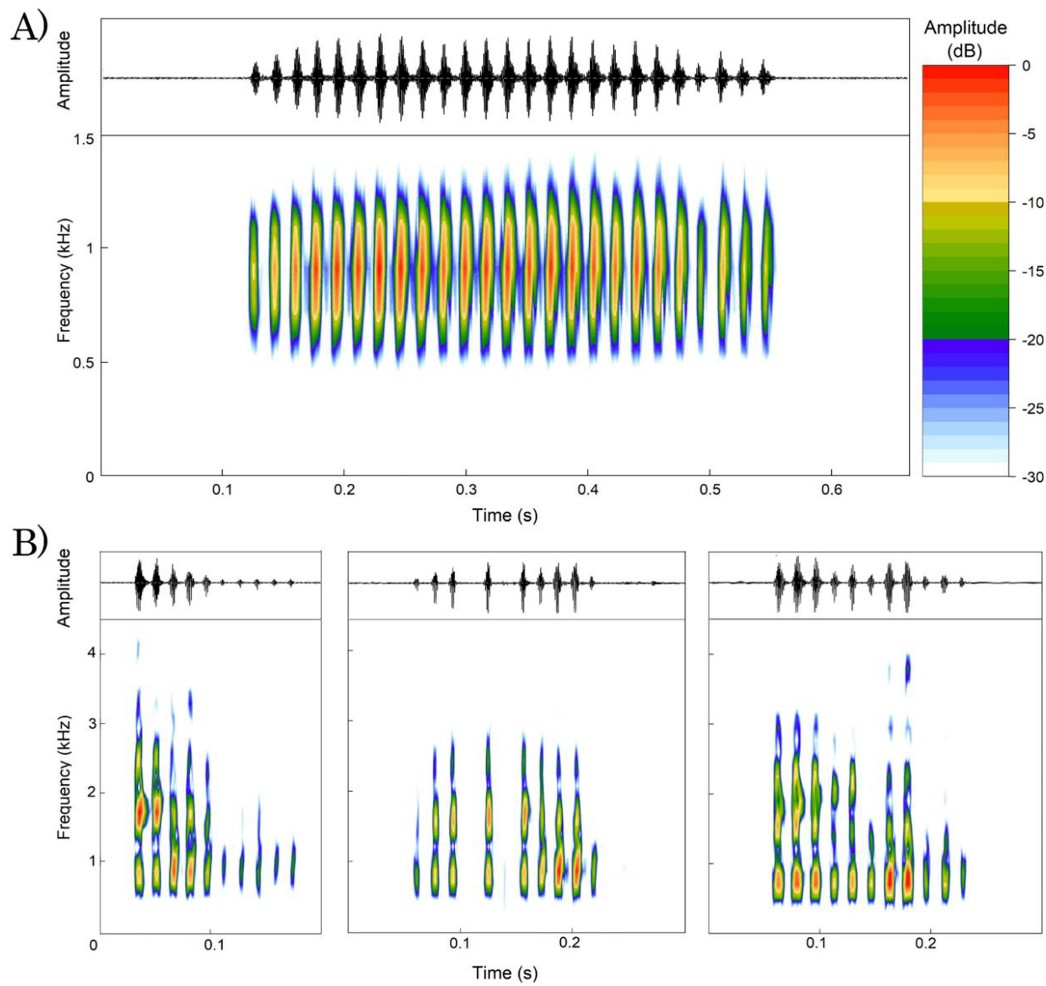

**Figure 4 Oscillogram and spectrogram of *Proceratophrys paviotii* recorded at Santa Teresa, State of Espírito Santo, Southeastern Brazil.** (A) Advertisement call; and (B) three release calls showing different patterns of amplitude and frequency modulation. The scale of amplitudes is decibels (dB), where colors represent different amplitude levels. The scale ranges from 0 to 30 dB, with warm colors indicating lower amplitudes and cool colors indicating higher amplitudes. The colors are distributed as follows: Red for 0 dB, Orange for 5 dB, Yellow for 10 dB, Light Green for 15 dB, Dark Blue for 20 dB, Light Blue for 25 dB, and White for 30 dB.                                               

**Table 4 Intra- and interindividual coefficients of variation of the advertisement calls of *Proceratophrys paviotii* from the Municipality of Santa Teresa, State of Espírito Santo, Southeast Brazil.** *Static variables.

| Acoustic signal | CVintra | CVinter | CVinter/CVintra |
|---|---|---|---|
| Note duration | 9.3 | 14.6 | 1.5 |
| Peak frequency | 1.6* | 3.8* | 2.3 |
| 5% frequency | 2.8* | 5.9* | 2.1 |
| 95% frequency | 3.9* | 5.4* | 1.4 |
| Pulses/call | 8.6 | 16.8 | 2.0 |
| Pulse rate | 2.1* | 9.7 | 4.4 |

*P. moehringi* by its higher peak of frequency. For detailed comparison and literature sources, see Table 3.

## DISCUSSION

Data gathered by citizen scientists were crucial in discovering new populations of *Proceratophrys paviotii*, enabling valuable systematic evaluations. Our phylogenetic analysis grouped *Proceratophrys paviotii* with *P. cururu*, *P. renalis*, and *P. laticeps* (PP = 0.91) in a clade, and *P. appendiculata*, *P. belzebul*, and *P. tupinamba* in a sister clade (PP = 1.00). Sequence divergence ranged from 2.2% to 9.1%. Advertisement calls consisted of single-note calls lasting 0.26–0.58 s with 17–41 pulses emitted at a rate of 54.19–77.49 pulses/s, and peak of frequency at 775.19–947.46 Hz. Additionally, the analysis of 55 release calls from a single male showed durations of 0.04–0.43 s with 2–13 pulses emitted at a rate of 21.17–81.58 pulses/s, and peak of frequency at 689.1–1,722.6 Hz. Variation analysis indicated dynamic traits.

The genus *Proceratophrys* currently encompasses 43 species with occurrences in Brazil, Argentina, Paraguay, and possibly in Bolivia (*Frost, 2024*). Historically, most *Proceratophrys* species have been arranged into four morphological groups, currently considered non-monophyletic: the *P. appendiculata* complex, the *P. bigibbosa* group, the *P. boiei* complex, and the *P. cristiceps* group (*Izecksohn, Cruz & Peixoto, 1998*; *Giaretta, Bernarde & Kokubum, 2000*; *Kwet & Faivovich, 2001*; *Prado & Pombal, 2008*). We did not recover the monophyly of these phenotypic arrangements in accordance with previous studies (*Mângia et al., 2020*; *Santana et al., 2021a*, *2021b*; *Mângia et al., 2022*).

Among anurans, genetic distance of 3% (16S rDNA) between taxa is commonly used as threshold to consider a lineage as candidate species (*Fouquet et al., 2007*) (*i.e.*, genetic distance below this threshold is commonly observed intraspecifically). Among *Proceratophrys* spp., four species showed genetic distance below this threshold compared to *P. paviotii*: *P. appendiculata*, *P. cururu*, *P. laticeps*, and *P. tupinamba*. In spite of this low genetic distance, we reinforce that *P. paviotii* can be easily diagnosed from these species based on the following combination of characters: presence of palpebral appendages and lack of rostral appendage (*Prado & Pombal, 2008*). In addition, other *Proceratophrys* were proven to be morphological, genetic, and geographically distinguished, and have been previously described or validated with values lower than 3% of genetic distance (*Dias et al., 2013*; *Magalhães et al., 2020*; *Santana et al., 2021a*).

Bioacoustic characters have commonly shown to be relevant in diagnosing species of *Proceratophrys* (*e.g.*, *Cruz & Napoli, 2010*; *Martins & Giaretta, 2011*; *Godinho et al., 2013*; *Mângia et al., 2014*, *2018*, *2022*). However, 12 species of *Proceratophrys* still lack call description. Among the 31 species of *Proceratophrys* with described vocalization, *P. paviotii* stands out due to its notably subsampling. The description provided by *Cruz, Prado & Izecksohn (2005)* is based on seven calls emitted by a single male. Such a sample limitation certainly makes taxonomic comparisons questionable because it does not properly encompass intra-specific variation. Herein we increased samples of both the number of analyzed calls (seven calls in *Cruz, Prado & Izecksohn (2005)* and 107 in the present study) and recorded males (one in *Cruz, Prado & Izecksohn (2005)* and 13 in the

present study). Nevertheless, the advertisement call of *P. paviotii* distinguishes it from 22 out of 30 *Proceratophrys* spp. (Table 3). Given that the advertisement call of *P. paviotii* differentiates it from the majority of species within the genus based on various parameters, such as note duration, pulse emission rate per call, and dominant frequency, we reinforce the importance of using bioacoustics characters in diagnosing *Proceratophrys* species.

Spectral parameters and pulse rate are commonly considered static in amphibians while temporal parameters are classified as dynamic (*Gerhardt, 1991*; *Bee et al., 2001*). Similar pattern was observed for *Proceratophrys paviotii*. The ratio between CVinter and CVintra observed indicates that there is greater variation between males than the individual variation itself, as also observed in other species of anurans (*e.g.*, *Morais et al., 2012*; *Forti, Lingnau & Bertoluci, 2017*). Furthermore, all advertisement call properties examined in this study exhibited greater variability inter-individuals than intra-individuals, suggesting the possibility of individual recognition (*Bee et al., 2001*; *Pettitt, Bourne & Bee, 2013*; *Moser et al., 2022*).

A taxon is classified as Near Threatened (NT) when is close to or likely to qualifying for a threatened category in the near future (*IUCN, 2024*). However, this does not appear to apply to *P. paviotii*. In several studies conducted across its distribution range in the central and northern parts of the State of Espírito Santo (Fig. 1) (*Cruz, Prado & Izecksohn, 2005*; *Prado & Pombal, 2008*; *Almeida, Gasparini & Peloso, 2011*; *Peres & Simon, 2011*; *Zocca, Tonini & Ferreira, 2014*; *Silva et al., 2018*; *Ferreira et al., 2019a*, *2019b*), the species has been consistently observed within protected areas (*Cruz, Prado & Izecksohn, 2005*), as well as in disturbed areas such as coffee plantations (see *Prado & Pombal (2008)* comment on *Teixeira & Coutinho (2002)*), and even urban areas (present study). Thus, our results suggest that *Proceratophrys paviotii* should be classified as Least Concern regarding its conservation status in future evaluations. Additionally, we emphasize that, despite suggesting the classification as Least Concern, assessing the species under the Green Status framework can aid in monitoring the effectiveness of the conservation actions presented herein. This methodology aids in tracking how close the species is to being "Fully Recovered" and in devising more efficient conservation strategies (*IUCN, 2024*).

The *Cantoria de Quintal* CS project stands as a pioneering initiative in Brazil, with specific focus on frog vocalizations. Although the project is limited to the State of Espírito Santo, it has managed to gather calls from more than 40 anuran species, including the Near Threatened *Proceratophrys paviotii*. Our study strongly supports the notion that Citizen Science approaches can yield invaluable information concerning species' geographic distribution and taxonomy. The data obtained through this participatory approach have provided important insights for conservation efforts, underscoring the potential significance of involving the public in scientific research and conservation initiatives.

Despite the many new studies using Citizen Science, this approach for Neotropical amphibians is still incipient. The approach used in "Cantoria de Quintal" was paramount in discovering new individuals of *Proceratophrys paviotii*. The genus *Proceratophrys* includes opportunistic breeders that call only after heavy rains (*Prado & Pombal, 2005*; *Godinho et al., 2013*; *Mângia et al., 2014*; *Malagoli, Mângia & Haddad, 2016*), and

many species in the genus have been described based on limited type series due its rarity (*e.g.*, *Prado & Pombal, 2008*; *Mângia et al., 2014*). Therefore, initiatives such as "Cantoria de Quintal," which utilize a contributive Citizen Science approach, should be encouraged in future studies, as they have demonstrated their effectiveness in targeting rare or explosively breeding species.

## CONCLUSIONS

We conducted the first molecular analysis to determine the phylogenetic position and species delimitation of the Near Threatened *Proceratophrys paviotii*. Our findings did not support the monophyly of phenotypic arrangements within *Proceratophrys*, consistent with previous studies. The 16S tree confirmed *P. paviotii* as a valid species, grouping it with *P. cururu*, *P. renalis*, and *P. laticeps* in a clade. The advertisement call of *P. paviotii* distinguishes it from 22 other species within the genus based on various parameters, highlighting the importance of bioacoustic characters in diagnosing *Proceratophrys* species.

Although Near Threatened species are close to qualifying for higher risk categories, *P. paviotii* has been observed in both protected and disturbed areas, including coffee plantations and urban regions. Our results suggest that *P. paviotii* should be classified as Least Concern in future conservation assessments. We also emphasize the value of Citizen Science in gathering information on species distribution, taxonomy, and conservation, contributing to the scientific knowledge and conservation of *Proceratophrys paviotii*.

## ACKNOWLEDGEMENTS

We would like to thank Joziane da Silva Broseguini, Alba Lívia Tallon Bozi, Kesia Faian, Emerson Araújo de Miranda, Viviane Vieira Lopes, Lucelia Barth, Julia Meirelles, Ruan Pablo Ramos França, and Luiza Loss Zanette for sending audio files through the Cantoria de Quintal citizen science project. Larissa Lacerda Moraes for kindly building the map's plate. We thank the editor Andrew Gregory, Ibere Machado and two anonymous reviewers for the critic revision of the manuscript.

### Funding

This study had financial support from the National Council for Scientific and Technological Development (CNPq, Programa de Capacitação Institucional–PCI/INMA) of the Brazilian Ministry of Science, Technology and Innovation (MCTI) (#301349/2023-1, #317325/2023-0 and Ordinance n° 143, February 16, 2023). Diego J Santana was supported by CNPq (Conselho Nacional de Desenvolvimento Científico e Tecnológico) research fellowship (309420/2020-2). Carla Guimarães (CAPES, Finance Code 001) and Alan Araujo (CAPES 88882.347126/2019-01) were supported by the Coordenação Aperfeiçoamento de Pessoal de Nível Superior for Ph.D. fellowships. The APC for this study was financed by the Coordenação de Aperfeiçoamento de Pessoal de Nível

Superior - Brasil (CAPES) - Finance Code 001. The funders had no role in study design, data collection and analysis, decision to publish, or preparation of the manuscript.

## Grant Disclosures

The following grant information was disclosed by the authors:
National Council for Scientific and Technological Development (CNPq, Programa de Capacitação Institucional–PCI/INMA) of the Brazilian Ministry of Science, Technology and Innovation (MCTI): #301349/2023-1, #317325/2023-0 and 143.
CNPq (Conselho Nacional de Desenvolvimento Científico e Tecnológico) Research Fellowship: 309420/2020-2.
Coordenação Aperfeiçoamento de Pessoal de Nível Superior (CAPES): 001 and 88882.347126/2019-01.

## Competing Interests

The authors declare that they have no competing interests.

## Author Contributions

- João Victor Andrade Lacerda conceived and designed the experiments, performed the experiments, analyzed the data, prepared figures and/or tables, authored or reviewed drafts of the article, and approved the final draft.
- Diego J. Santana conceived and designed the experiments, performed the experiments, analyzed the data, prepared figures and/or tables, authored or reviewed drafts of the article, and approved the final draft.
- Carla Guimarães conceived and designed the experiments, performed the experiments, analyzed the data, prepared figures and/or tables, authored or reviewed drafts of the article, and approved the final draft.
- Alice Zanoni dos Santos performed the experiments, analyzed the data, authored or reviewed drafts of the article, and approved the final draft.
- Alan P. Araujo conceived and designed the experiments, performed the experiments, analyzed the data, prepared figures and/or tables, authored or reviewed drafts of the article, and approved the final draft.
- Natalia Pirani Ghilardi-Lopes conceived and designed the experiments, analyzed the data, authored or reviewed drafts of the article, and approved the final draft.
- Sarah Mângia conceived and designed the experiments, performed the experiments, analyzed the data, prepared figures and/or tables, authored or reviewed drafts of the article, and approved the final draft.

## Animal Ethics

The following information was supplied relating to ethical approvals (*i.e.*, approving body and any reference numbers):

The Instituto Chico Mendes de Conservação da Biodiversidade (ICMBio #63575-5) and animal research ethics committee of the Universidade de Vila Velha (CEUAUVV #491-2018) provided sampling permits.

## DNA Deposition

The following information was supplied regarding the deposition of DNA sequences:

The sequences generated from this study are available at GenBank: PP442191–PP442193 and in the Supplemental Files.

Existing sequences used are available at GenBank: DQ283097, FJ685684, FJ685685, JX564880, KP295642, FJ685686, FJ685687, FJ685688, AY843704, FJ685694, KF214151, KF214152, KX858852, KX858853, KX858854, DQ283039, FJ685691, KP295643, KF214154, KF214155, KF214156, FJ685692, MG798659, MG798660, JN814630, JN814653, JN814662, JN814592, JN814620, JN814648, JN814586, JN814612, JN814660, KU495472, KX858855, FJ685695, MF953400, MF953401, FJ685696, KU495477, KU495478, FJ685697, KU495479, FJ685699, KF214142, KF214147, KF214157, KU495483, MW916089, MW916090, MW916088, MW889930, MW889928, MW889929, FJ685698, KF214143, KF214140, KF214149, JX982965, JX982966, FJ685689, MT196403, KF214144, KF214148, JX982967, JX982968, FJ685700, JN814584, MT196397, MT196399, FJ685701, KU495473, LTJV180, MHO117, MHO290, MHO309, KF214158, KF214159, KF214160, MT537176, MT537177, FJ685682.

## Data Availability

The audio files are available at Fonoteca Neotropical Jacques Vielliard (FNJV), search only using the number, not including the letters: FNJV_0060377, FNJV_0060378, FNJV_0060379, FNJV_0060380, FNJV_0060381, FNJV_0060382, FNJV_0060383, FNJV_0060384, FNJV_0060385, FNJV_0060386, FNJV_0060387, FNJV_0060388, FNJV_0060389, FNJV_0060390.

## Supplemental Information

Supplemental information for this article can be found online at http://dx.doi.org/10.7717/peerj.17990#supplemental-information.

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
