# Peer review of "Combining citizen science, phylogenetics, and bioacoustics to inform taxonomy and conservation of the Near Threatened Proceratophrys paviotii (Anura, Odontophrynidae)"

_PeerJ, doi:10.7717/peerj.17990_

## Round 0.1 · original submission · Minor Revisions

Dear Authors,

I have now received 4 very good reviews of your paper. Based on these and my own review of your paper I am recommending a MINOR REVISION.
The reviewers all noted that the paper was well written and utilized a robust methodology. The team also noted that with the rise in interest in the use of citizen science, your paper will likely be of high relevance and interest to the scientific community.

In terms of your revision, two of the reviewers, and myself, noted that your management recommendations seem to exceed your data. For example, it is unclear how the revised taxonomy and occupancy modeling can change the status of a species. Your study, as presented, was not a systematic survey of the species range-wide, nor did your study address demographic performance or population viability at any occupied region, only presence. It is possible that all occupied patches would no longer be occupied in e.g., 5-years and presence at those sites is ephemeral. In your revision please make sure to adequately temper or caveat any management recommendations based on your data.

The reviewers also noted that while overall the paper was well-written and edited, there was still room for improvement. Of the four reviewers, reviewer 2 was the only native English speaker, and they note that the paper could use a review by a native speaker. The reviewer did provide a set of robust and annotated edits to this effect. Reviewer 4 also provided some editorial comments directed at improving the clarity of the English in the paper. Finally, three of the reviewers have asked that you reorder and revise the methods section for clarity and have provided edits to help in that regard.

Thank you very much for your work and your attention to these details in your revision. I look forward to reading the revised manuscript.

Sincerely,
Andy Gregory.

·

Basic reporting

The manuscript demonstrates clear and professional English language usage, providing an unambiguous presentation of the research findings and their significance within the context of the conservation of amphibians.
Throughout the manuscript, the introduction and background sections were supported by well-referenced updated literature. The figures and raw data have been thoroughly verified and found to be accurate, meeting the criteria for Basic Reporting. So, the manuscript is appropriately structured content in adherence to PeerJ standards, ensuring clarity and relevance.

Experimental design

The experimental design of the paper adheres to the journal's Aims and Scope, offering original primary research that addresses a pertinent and well-defined research question, highlighting its relevance within the field. The investigation is conducted rigorously, maintaining high technical and ethical standards throughout the study, thereby ensuring the reliability and validity of the findings. Detailed descriptions of the methods provide clarity and transparency, enabling replication and further contributing to the advancement of scientific knowledge in the field.

Validity of the findings

The validity of the findings is upheld through robust data provision, ensuring statistical soundness and control, and promoting meaningful replication where the rationale and benefit to the literature are clearly articulated. Conclusions are tightly linked to the original research question, staying within the boundaries of supporting results and refraining from overstating implications.

Additional comments

Dear Editor and Authors,

After conducting a thorough review of the manuscript, I am delighted to recommend it for acceptance for publication. The manuscript adheres to the journal's guidelines impeccably and makes significant contributions to the field. I eagerly anticipate its publication and dissemination within the scientific community.

Upon closer examination, the manuscript presents invaluable information essential for the conservation of the species. With each new scientific discovery, Red List assessments aim to classify species with the utmost accuracy, updating classifications based on the latest data. Furthermore, it is worth highlighting that, despite proposing classification as Least Concern, evaluating the species under the Green Status framework can enhance monitoring of the effectiveness of the conservation actions presented here.

I would also like to suggest that the authors consider outlining the "next steps" for the conservation of the species, should they wish to include it. This addition would offer valuable insights into potential future directions and strategies for conservation efforts. If agreeable to the authors, I recommend adding a statement along these lines: "Additionally, we emphasize that, despite suggesting the classification as Least Concern, assessing the species under the Green Status framework can aid in monitoring the effectiveness of the conservation actions presented herein. This methodology aids in tracking how close the species is to being "Fully Recovered" and in devising more efficient conservation strategies (IUCN, 2024)."

Ref: https://portals.iucn.org/library/node/49511

Reviewer 2 ·

Basic reporting

The English language should be improved to ensure that an international audience can clearly understand your text. I suggest you have a colleague who is proficient in English and familiar with the subject matter review your manuscript, or contact a professional editing service.
Overall this is an interesting study that uses multiple methods to validate the findings. I would recommend rewriting/reorganizing the introduction and discussion to provide a better background on why this study is interesting and important. Please see attached PDF for further comments.

Experimental design

I appreciate the amount of work that went into using multiple methodologies to determine the occurrence and taxonomy of P. paviotii. Your research objectives are clear and well defined. For the most part your methods are well described, but I think it would be beneficial to include a paragraph or two about the study area. Please see attached PDF for further comments.

Validity of the findings

This is a robust study that used bioacoustics and phylogenetics to determine the taxonomy of P. paviotii. However, I think making recommendations about the species’ conservation status is outside the scope of the current study. Please see attached PDF for further comments.

Additional comments

Please see the attached PDF.

Annotated reviews are not available for download in order to protect the identity of reviewers who chose to remain anonymous.

Reviewer 3 ·

Basic reporting

Dear Editor and Authors,

I have carefully read the article entitled "On the Near Threatened Proceratophrys paviotii (Anura, Odontophrynidae): combining taxonomy to bioacoustic, citizen science, and conservation" by João Victor Andrade de Lacerda and coauthors.

I think that this article presents a small, but valuable contribution to our knowledge of variation, ecology and taxonomy of Proceratophrys. I have only a few critical remarks and they are rather minor.

In general, the article is well-written. My main criticism is that in my opinion the 'Materials & Methods' section is rather chaotic. For example, the information on what tissue samples were collected is given in the subsection 'Occurrence data by specialists' (in my opinion, it would fit better in 'Phylogenetic inference and genetic distances' where the methods of DNA extraction from tissue is described), information on what files were used for bioacoustic analysis is given in 'Occurrence data through Citizen Science' rather than 'Bioacoustics' etc. I recommend to arrange these information in more logical order.

The text is comprehensible, but there are a few phrases that sound awkward to me (a few examples are given below, but I'm not a native speaker and don't feel comfortable in making thorough language assessment).

Figures are ok in my opinion, but Figure 2 (the phylogenetic tree) is poorly legible, at least in my review PDF (and the species name is misspelled).

Experimental design

The research is original and within the scope of the journal. I don't have any critical comments here.

Validity of the findings

The conclusions seem to be valid based on the provided data. I don't have any critical comments here.

Additional comments

Specific suggestions:

line 3 (and elsewhere)
I think that 'neotropical' should be capitalised.

line 40: "In anurans, advertisement calls are inheritable and usually species-specific (Duellman & Trueb, 1994; Wells, 2007)."
Replace "inheritable" with "heritable" (or equivalent, like "genetically determined").

line 51: "...using smartphones (specially, but not only audio files)"
"Not only audio files" means that both audio and video files were accepted? If so, I would indicate it more clearly.

line 63: encompassing --> encompass

line 65: "began a taxonomic integrative investigation on the Proceratophrys paviotii taxonomy"
'Taxonomic investigation on taxonomy' sounds like a tautology to me. I suggest to change it to 'integrative investigation on Proceratophrys paviotii taxonomy' or 'integrative taxonomy investigation on Proceratophrys paviotii'.

line 85: Indicate what tissue samples.

lines 131-132: "Vocalization recordings were deposited at Fonoteca Neotropical Jacques Vielliard (FNJV; https://www2.ib.unicamp.br/fnjv/)."
I presume these files will be publicly accessible when the article is published? (at present, I couldn't locate them in the database).

lines 161-163: "Because of recording quality, only files FNJV 60388 and FNJV 60389-60390 were included in the bioacoustical analysis. The other files were considered for mapping the species geographical distribution only."
I think this should be placed in the previous section ('Bioacoustics').

line 231: I think that these groups were initially thought to be monophyletic, at least potentially, so I suggest rephrasing the sentence into, e.g., "...have historically been arranged into four morphological groups, currently cinsidered non-monophyletic".

lines 261-275
In this paragraph, you just list the differences between P. paviotii and other species, you do not discuss them. I think that this paragraph is more suitable to be in the 'Results' (section on bioacoustics) rather than in 'Discussion'.

lines 304-322: This paragraph seems to be mostly 'copy-paste' from 'Discussion'. Could you please rephrase it at least slightly?

lines 321-322: I suggest to remove the last sentence.

lines 378, 429-430, 458: Journal names should not be abbreviated.

lines 394-395: This DOI refers to book reviews, not the book itself.

line 435: Add journal title.

line 477: Please give final citation (24:181-185).

line 562: This citation is unclear to me.

In the references, there are many words that should be capitalised or italicised (or both), but are not (e.g., lines 330-331: "state of espírito santo, southeastern brazil", line 601: "proceratophrys miranda-ribeiro, 1920")

Figure 1: Italicise Proceratophrys paviotii

Figure 3: pavitotii --> paviotii (in the label), paviotti --> paviotii (in the figure)

Tables 2: "ofProceratophrysspecies" (twice) - insert spaces

Reviewer 4 ·

Basic reporting

The manuscript is very well written in clear and proper English language. All the figures, tables and references are adequate and necessary. The publication is completely “self contained” and original.

Experimental design

The manuscript is original, with consistent results supporting all the conclusions and the Discussion. The research question is very well defined, is relevant and meaningful. The authors could present the importance of the citizen science to the advance of the knowledge in taxonomy, systematics, natural history and conservation of a poorly known (until now) frog species.

The methods are properly presented and all of them is replicable.

Validity of the findings

In a first view I thought the manuscript could be of reduced impact. However, the authors did an excellent job arguing in favor of citizen science and presenting the gaps in the knowledge on the systematics and the apparently mistaken conservation status of Proceratophrys paviotti. They changed my first impression, and, in the end of my reading, I am convinced the results presented has relevant impact and bring noteworthy novelties. The data are robust and properly analyzed. The conclusions are all supported by the results presented and well postulated.

Additional comments

I would like to commend the authors for their job in presenting the importance and applicability of the citizen science and how very simplistic collaborations from non formal scientists. The authors presented a simple, but very important contribution to the systematics, taxonomy, natural history and conservation of the genus Proceratophrys.

I am completely convinced that the manuscript fits in the criteria and scope of PeerJ and I would like to see it published.

I made few comments and corrections directly on the attached PDF file.

Annotated reviews are not available for download in order to protect the identity of reviewers who chose to remain anonymous.

---

## Round 0.2 · accepted · Accept

I apologize for the delay in getting this decision to you. After additional review, I am happy to inform you that the reviewers and I have all concluded that the paper is nearly ready for publication. There are a few minor grammatical and stylistic suggestions made by each of the reviewers, but these do not represent changes in content or scope of the paper. Therefore I recommending that your paper be accepted to Peer-J.

Reviewer 2 ·

Basic reporting

The manuscript looks great! I appreciate the time you took to incorporate the reviewers recommendations. I still have some very minor edits that Ill put in the additional comments section, mainly just on grammar.

Experimental design

No comment

Validity of the findings

No comment

Additional comments

The paragraph on lines 55-61 seems incomplete?
Line 62 – Change records to recordings.
Line 217/218 – the ethical treatment of animals or vertebrates would sound better.
Line 242/243 – Calls had an ascendant amplitude modulation on their beginning and descendant amplitude modulation on their ending.
Line 308 – notably to notable
Line 319 – A similar pattern
Line 324 – greater variability between inter-individuals than intra-individuals

Reviewer 3 ·

Basic reporting

Dear Editor and Authors,

I have reviewed the revised version of the article by João Victor Andrade de Lacerda and coauthors. This is a nice article and I hope it will be published soon.

My main criticism of the first version was about rather chaotic organisation of the methods and I'm glad to see the article improved in this respect. I have only a few suggestions now and they are very easy to implement.

Although I'm not a native speaker and can't do a thorough language evaluation, I'd like to suggest a few changes to improve the language (please see 'additional comments' below).

Experimental design

The research is original and within the scope of the journal. I don't have any critical comments here.

Validity of the findings

The conclusions seem to be valid based on the provided data. I don't have any critical comments here (I refrain from judging the correctness of proposals on the conservation status).

Additional comments

line 19: basel → based

line 42: "Critically Endangered (CR), Vulnerable (VU), or Endangered (EN)"
That's a weird sequence. I suggest to put it in the order of either CR, EN, VU or VU, EN, CR.

lines 157-158: "We identified each species using different colors in the haplotype network."
"Identified" or "highlighted"?

lines 277-285: This is a brief summary of the results. In my opinion, it doesn't have to be repeated here.

lines 301-302: "were proven to be morphological, genetic, and geographically"
Please change to "were shown to be morphologically, genetically, and geographically"

lines 304-305: "Bioacoustic characters have commonly shown to be relevant in diagnosing species of Proceratophrys"
Please change to "Bioacoustic characters have often been shown to be relevant in diagnosing species of Proceratophrys"

line 308: notably → notable

Figure 3: Something weird happened with the caption of this figure. There's unneeded title of the article inserted in both the title and legend of the figure. There are also some typos ("ûgure"). Please check the caption carefully. Also, the species name on the haplotype network is misspelled as "P. paviotti".